# The Effects of Listening to Music on Postural Balance in Middle-Aged Women

**DOI:** 10.3390/s24010202

**Published:** 2023-12-29

**Authors:** Fatma Ben Waer, Sonia Sahli, Cristina Ioana Alexe, Maria Cristina Man, Dan Iulian Alexe, Lucian Ovidiu Burchel

**Affiliations:** 1Research Laboratory Education, Motricity, Sport and Health, LR19JS01, High Institute of Sport and Physical Education of Sfax, University of Sfax, Sfax 3000, Tunisia; fatmaelwaer123@gmail.com (F.B.W.); sonia.sahli.isseps@gmail.com (S.S.); 2Department of Physical Education and Sports Performance, Faculty of Movement, Sports and Health Sciences, “Vasile Alecsandri” University of Bacau, 600115 Bacau, Romania; 3Department of Physical Education, 1 Decembrie 1918 University of Alba Iulia, 510009 Alba Iulia, Romania; 4Department of Physical and Occupational Therapy, Faculty of Movement, Sports and Health Sciences, “Vasile Alecsandri” University of Bacău, 600115 Bacau, Romania; alexedaniulian@ub.ro; 5Department of Environmental Sciences, Physics, Physical Education and Sports, Faculty of Sciences, Lucian Blaga University of Sibiu, 550024 Sibiu, Romania; lucian.burchel@frf.ro

**Keywords:** music, postural balance, women

## Abstract

Listening to music has been found to influence postural balance in both healthy participants and certain patients, whereas no study investigates such effects among healthy middle-aged women. Thus, this study aimed to investigate the effect of music on postural balance in middle-aged women. Twenty-six healthy women aged between 50 and 55 years participated in this study. A stabilometric platform was used to assess their postural balance by recording the mean center of pressure velocity (VmCOP) in the eyes-opened (OE) and -closed (EC) conditions on both firm and foam surfaces. Our results showed that listening to an excerpt of Mozart’s Jupiter significantly decreased the VmCOP values in two sensory conditions (firm surface/EO: (*p* < 0.01; 95% CI: 0.27 to 2.22); foam surface/EC: (*p* < 0.001; 95% CI: 0.48 to 2.44)), but not in the other two conditions (firm surface/EC and foam surface/EO). We concluded that listening to Mozart’s symphony improved postural performance in middle-aged women, even in challenged postural conditions. These enhancements could offer great potential for everyday functioning.

## 1. Introduction

Postural balance is an essential aspect of quality of life in the general population. Generally, postural control enables an active, functional, independent, and safer life. It is achieved by the integration of visual, somatosensory, and vestibular information through the central nervous system (CNS) to contract muscles adequately for maintaining balance [1]. Balance loss has potential direct consequences on the personal and functional independence that are related to an individual’s ability to carry out daily activities [2]. Strong evidence suggests that postural balance is negatively affected by aging [3,4], with substantial changes in postural control appearing in middle age [5], mainly among women. Indeed, it has been shown that there is a significant decline in women’s postural balance by the fifth and sixth decades of life [6]. Particularly, a negative alteration in mediolateral postural balance for women between the ages of 40 and 60 years was observed [7]. Consequently, such postural declines present a high risk of falls and fall-related injuries [8]. In combination, falls have been reported to be the third leading cause of unintentional injury mortality among those aged 45–64 years [9,10] and are more common in women compared to men [11]. Fall-related injuries in older women are more severe, and their costs are estimated to be 2–3 times higher than those of fall injuries in older men [12], and the percentage of women aged 51–60 who fall is higher than in any other age group (61–70, 71–80, and 81+ years) [11]. These falls can have a profound impact on the quality of life for women, resulting in significant morbidity and mortality rates, which leads to higher healthcare costs [13]. These falls impose dramatic consequences on the individuals who experience them. They not only result in physical impairments but also give rise to psychological and social issues. The interplay of these three consequences exacerbates functional decline, ultimately contributing to morbidity [14]. As a result, falls, whether traumatic or not, have a substantial effect on the overall quality of life [15].

Bearing in mind that the health of middle-aged women has become a major public health concern worldwide, and that balance disorders are not only an issue for older adults but also for middle-aged women, we became concerned with this subgroup of women because of their susceptivity to postural balance problems [16]. It seems interesting to identify strategies that target middle-aged women to have a holistic impact on postural sway-related injuries and disability. Yet, middle age implies unbearable challenges in women’s lives such as worries about growing old, marital stress, retirement, widowhood, or family deaths. Women at this stage of life are often referred as the sandwich generation, since they are often providing financial support and care for their family (i.e., children and aging parents) [17,18]. Owing to all these responsibilities, these women have multiple barriers to being regularly physically active, such as not being able or committed to carrying out physical activity during leisure time [19].

Interestingly, music is a very important component of everyday life and an essential part of leisure time for women, as it is considered one of the most satisfying and pleasurable everyday activities across different times and cultures [20,21]. Indeed, listening to music with personal listening devices like MP3 players, smartphones, or other portal music players while performing various activities of daily living is very common in people of different ages [22,23].

Music has been widely acknowledged as a cheap, safe, and effective intervention that has gained increasing recognition as an effective tool to reduce pain and anxiety and enhance mood and neuromotor and cognitive functioning [24,25]. Listening to music, indeed, captivates attention, elevates spirits, triggers a range of emotions, enhances motivation, regulates mood, evokes memories, increases work output, heightens arousal, induces higher states of functioning, and promotes motor coordination and rhythmic movements [26,27]. 

It functions as a specific stimulus to obtain motor and emotional responses by stimulation of different sensory pathways [28]. Various investigations have shown that listening to music induces straighter posture control, stronger and more symmetric movements, and better awareness of themselves and their environment in patients [29] through stimulating the motor-related structures of the brain, mainly the lateral premotor, supplementary motor, and somatomotor areas [30]. For that, many studies [23,31,32,33,34,35,36] have examined postural responses under music stimuli to investigate their associations. In these studies, a beneficial effect of music listening (such us relaxing music, the Bluebell Polka, and classical music) on postural balance has been revealed in different populations (in healthy subjects [31,32], young adults [23] and older adults [33], and in patients [34,35,36]). 

In contrast, some studies reported no significant effects of listening to music on postural performance, suggesting that the benefits of music may depend on the genre [37]. Yet, a previous study examined the effects of listening to different genres of music and found that different genres of music including rock, classical, and pop did not influence postural balance in young adults [23]. These authors explained these results by the fact that their participants were young, having a high capacity to easily compensate for balance changes [23]. However, it is commonly known that classical music shows the clearest beneficial effects on health [38]. Also, Gasenzer et al. [39] indicated that listening to classical music, mainly Mozart, which is known for its clarity, harmony, and transparency, has an impact on how the brain works and how people behave, such as by enhancing mood and cognitive and physical skills [39]. Importantly, it has been found that the positive effects of listening to Mozart’s Jupiter significantly improved both static [31] and dynamic balance [32], indicating the use of this specific type of music in the rehabilitative protocols to prevent falls. In this context, Mozart’s music pieces have attracted the most interest from clinicians and scientists compared to music pieces by other composers [40,41]. 

Despite all these beneficial effects of music, to date, no study has evaluated the effect of listening to music on postural balance in middle-aged women. The mechanism of this effect of music on the postural performance, mainly static balance, is still unknown. Based on all the considerations mentioned above and the need for further research on the different subcategories of music interventions, this research aims to explore the acute effects of listening to music on postural balance performance in middle-aged women. 

Determining whether their postural balance would be improved or impaired when listening to Mozart’s music would be helpful in lowering the risk of falls and protecting these women’s health and well-being. Therefore, we conducted a randomized, crossover trial to test the hypothesis that music by Mozart enhances the postural balance performance of middle-aged women. Considering that the force platform technique is one of the most commonly applied tools for assessing postural balance [42] and that posturography data have been shown to correlate with fall risk factors [43], we used a similar tool (a stabilometric platform) to assess the static balance of the participants under different sensory conditions. We hypothesized that Mozart’s music improves postural performance in these women, and that these effects depend on the sensory condition. Given that more attention needs to be paid to middle-aged women, as insight into postural disorders in this group is potentially important for delaying problems later in their life, methods such as musical interventions may be of interest. In doing so, we hope to highlight the explanatory mechanism of music listening on postural performance and areas requiring further research, as well as identify a useful new way of promoting postural balance and preventing the risk of falls.

### Paper Structure

This paper is organized into the following sections: Section 1 presents the introduction section and the rationale for the study. Section 2 provides the related works. Section 3 presents detailed materials and methods that describe the recruited participants, the study design, and the tools that were used. The results are addressed in Section 4 along with tables and figures showing the observed outcomes. Section 5 presents the discussion, in which we interpret and argue our main results and challenges that we have faced. Finally, the conclusion is presented in Section 6.

## 2. Related Works

Previous research has revealed that music can promote physical function, particularly gait and balance, in several populations, such as healthy individuals [23,31,32,33] and patients with visual impairments [36]. It could be considered an effective intervention to prevent falls. 

Indeed, Carrick et al. [32] investigated the effects of daily listening to different types of music on dynamic balance control in adults. Their findings suggest that certain types of music, such as Mozart’s or Nolwenn Leroy’s music, significantly improve dynamic posturographic scores, demonstrating that these music types have the potential to change postural stability and can be used as a fall prevention and rehabilitation method [32].

Hana et al. [36] investigated the acute effects of listening to Mozart’s Jupiter and preferred music on static balance under different sensory manipulations, using force plate posturography, among adolescents with visual impairment. The study found significant, positive, and large effects even in challenged postural conditions, suggesting that listening to these types of music may help minimize fall risks and related injuries and improve the quality of life for patients with visual impairment [36].

In accordance, Forti et al. [31] evaluated the influence of different types of music, including Mozart, Kohler, and the subjects’ favorite music, on healthy subjects standing on a stabilometric platform. These authors showed that listening to different types of music did not significantly change the stabilometric variables, except for Mozart’s music [31]. They reported that this particular type of music influences the postural balance strategy by decreasing the participants’ dependence on the visual sensory system while increasing their use of both vestibular and somatosensory inputs. These authors also asked for further investigation into how Mozart’s music can modify the sensory strategy in individuals with hearing or vision impairments [31].

## 3. Materials and Methods

### 3.1. Participants

From January to March 2023, in Romania, fifty-four middle-aged women from the general community were given handouts and explanations regarding the recruitment of study subjects, with direct advertisement through the authors’ family/friends and billboards. Then, those who agreed to participate in this study were selected. To calculate the required sample size, G* power software (version 3.1.9.2; Kiel University, Kiel, Germany) [44] was used. Values for alpha, power, correlations among repeated measures over group, and the non-sphericity correction (ε) were set at 0.05, 0.95, 0.5, and 1, respectively. While there is no published research investigating the connection between music and postural performance in middle-aged women, an a priori power estimation was conducted using a large effect size of Cohen’s f (0.4) to aid in clinical interpretability of the results. The required sample size was 12 participants to minimize the risk of Type II statistical error. Taking into account the possible drop-out of some participants, we recruited 26 participants, who met the inclusion criteria, to participate in our study. There were no drop-outs, and therefore, all the 26 participants were analyzed (Figure 1). The demographic and anthropometric characteristics of these women (age: 52.5 (±2.7) years; height: 1.56 (±0.8) m; mass: 78.3 (±6.5) kg) were collected from their medical files.

To ensure that all participants met all inclusion criteria, all of them were required to complete a health history questionnaire, the Blatt Kupperman Menopausal Index, Self-Assessment Questionnaire, fall risk questionnaire, and women’s health questionnaire. The inclusion criteria were as follows: (1) healthy, aged between 50 and 55 years, post-menopausal for at least 4 years, (2) classified as morning chronotype individuals with (3) a mild risk for falling. All women were physically independent with an absence of any physical or mental illness that could interfere with the assessment tests. Based on the information collected via their health history questionnaires, participants with vestibular or visual disorders, musculoskeletal or neurological diseases, grade III obesity, uncontrolled hypertension, ingestion of medication that can change balance (sedative and hypnotic agents), or with a history of cardiovascular, metabolic, renal, hepatic, or musculoskeletal disorders were excluded. This amounted to 28 women being excluded from the analysis.

The experimental protocol as well as the risks and benefits were explained to all participants. Following this explanation, all women gave their informed consent for inclusion before they participated in the study. The study was conducted in accordance with the Declaration of Helsinki, and the protocol was approved by the Ethics Committee of the Vasile Alecsandri University of Bacău (NR#5052/2/07.04.2023).

### 3.2. Study Design

A randomized, single-blinded, and counterbalanced crossover design was used to investigate the acute effects of listening to music on postural balance performance. Participants visited the laboratory on three different days, separated by 48 to 72 h, at the same time of day. 

In the first visit, we conducted a familiarization session 3 days before beginning the experimental protocol to eliminate the fear of new material. All tests were clearly verbally explained by the trained experimenters. During this session, participants were given a short trial (about 10 s) for each task to ensure that they were familiarized with the experimental protocol. The subsequent two visits were the testing sessions in which we evaluated participants’ postural balance in an upright bipedal stance under different sensory conditions: eyes-open (EO)/eyes-closed (EC) conditions on firm/foam surfaces conditions. The order of the conditions was randomly changed to minimize the order effect. Each experimental condition was conducted in two auditory conditions: no music (absence of auditory stimulus) and music (Mozart’s Jupiter) by wearing headphones. During the “no-music” period, the participants wore headphones that did not play any sound.

Three trials for each task condition (bipedal stand on firm surface/EO with no music, bipedal stand on firm surface/EC with no music, bipedal stand on foam surface/EO with no music, bipedal stand on foam surface/EC with no music, bipedal stand on firm surface/EO with listening to music, bipedal stand on firm surface/EC with listening to music, bipedal stand on foam surface/EO with listening to music, and bipedal stand on foam surface/EC with listening to music) were performed and then averaged for statistical analysis. One minute of rest between trials was observed to eliminate the fatigue effect. The trials of postural balance assessment while listening to music or under the no music condition were sequentially randomized into two blocks by an independent investigator. Non-music trials were performed in Block 1 and music trials were performed in Block 2 (Figure 1). A random number program (https://www.randomizer.org/ (accessed on 12 March 2023) was used to assign the order of trials for each participant (Figure 1). All trials in both Block 1 and 2 were also performed in a randomized order in order to avoid the influence of learning on outcomes. Before beginning, each participant declared that she correctly understood the test. All data were collected by two experienced experimenters. This study is a single-blinded trial, in which these two experimenters were not informed of the sound that the participant was listening to while performing a postural task.

The music by Mozart is Symphony No. 41 in C major, KV 551 Jupiter. It was played for 30 s, the time taken to assess postural balance while standing on the stabilometric platform, from the beginning at each trial. In all the task conditions, the same excerpt of Mozart’s music (Mozart’s Symphony No. 41 “Jupiter” in C Major, K. 551, starting with the first movement (“I. Allegro vivace”) from the opening of the symphony) was used. Once participants were comfortably positioned on the stabilometric platform, Mozart’s music was initiated simultaneously with the beginning of the CoP sway recordings. Any trial in which participants exhibited sudden movements or overreactions to the music’s onset was excluded from the analysis to prevent potential disturbances to the study’s outcomes. We downloaded Mozart’s music from YouTube in an MP3 form, a lossy audio codec with a bitrate of 320 kbps, presented in stereo. The music volume was set at 10/15 with an average 65 ± 5 decibel (dB), measured using an Android application (Sound Meter (ver. 1.6.5a)) in each test [23]. The same smartphone (Samsung Galaxy A30 (Suwon-si, Republic of Korea)) and its regular on-ear headphones were used for listening to music by each participant. The headphones feature 40 mm drivers, have a frequency response range of 20–28 Hz, an impedance of 32 ohms, and a sensitivity of 103 dB.

### 3.3. Postural Balance Assessment

Postural balance was measured using a stabilometric platform (posture Win©, Techno Concept^®^, Cereste, France; 40 Hz frequency, 12-bits A/D conversion) that records the center of pressure (CoP) sways with three strain gauges (Figure 2). The CoP motion was computed from the ground reaction forces and their associated torques. As participants oscillate during the upright standing postures with their body remaining relatively rigid, the reaction force applied to the body is almost constant, and so, the variations in the associated torque depend mainly on the CoP motion [45]. Thus, analyzing CoP motion amounted to analyzing muscular torques that controlled the body oscillations [45,46]. The women were asked to stand barefoot on the stabilometric platform, as immobile as possible on an upright bipedal posture with their arms comfortably placed downward at either side of the body; their bare feet were separated by an angle of 30°, and their heels placed 5 cm apart (Figure 2 and Figure 3). To maintain the same foot positions for all of the measurements, a plastic device provided with the platform was used only on the firm surface, as the device cannot be applied on an unstable surface (e.g., foam surface). Postural measurements were collected under two vision conditions. In the eyes-open condition, participants were instructed to keep their gaze horizontal, focusing on a visual target that was positioned 2 m away. In the eyes-closed condition, vision was eliminated by wearing a blindfold. For each of the vision conditions, women were tested under two surfaces conditions: firm surface and foam surface (Figure 3). The foam surface consisted of a foam block (466 mm length × 467 mm width × 134 mm height above ground) with a density of 21.3 kg/m^3^ and an elastic modulus of 20.900 N/m^2^ [47]. Each of these postural conditions (EO/EC in firm or foam surface) was conducted in two auditory conditions: no music (absence of auditory stimulus) and music (Mozart’s Jupiter) conditions (Figure 3 and Figure 4). The CoP mean velocity (VmCOP) was selected, as it is the most accurate form of sensory information used to stabilize posture during the quiet stance [48]. According to Salavati et al. (2009), the VmCOP (mm/s) formula is [49].
$$VmCOP=\frac{1}{T}\sum_{1}^{T}\sqrt{{\left(x_{t+1}-x_{t}\right)}^{2}+({y_{t+1}-y_{t})}^{2}}$$

Three trials were conducted in each experimental condition. The duration of each trial was 30 s, following the French Posturology Association norms. To cancel fatigue and/or learning effects, 60 s of rest was taken between trials. All experiments were assessed by the same raters who, during measurements, stayed near the participant for security without adducing any additional directions. The best postural balance is indicated by lower values of this parameter [50].

### 3.4. Statistical Analyses

The statistical analyses were carried out using the software Statistica 12 (StatSoft, Paris, France). The statistical significance value was set as α = 0.05. Values were expressed as means ± standard deviations (SD). The Shapiro–Wilk test revealed that data were normally distributed. The variance homogeneity was verified using the Levene test. A three-way ANOVA with repeated measures (2 vision × 2 surfaces × 2 auditory conditions) was used to determine the effects of the auditory conditions (no music vs. Mozart’s Symphony *No. 41*), vision (EO/EC), and surfaces (Firm surface and Foam surface) factors on the VmCOP values. For each significant main factor and interaction, a post hoc analysis was executed with the Bonferroni test [51], i.e., the multi-comparison alpha (α_MC_) is α_MC_ = α/12 = 0.00417. Effect sizes for the main and interaction effects were calculated using the partial eta squared (η^2^_p_) (small effect: 0.01 < η^2^_p_ < 0.06; medium effect: 0.06 < η^2^_p_ < 0.14; and large effect: η^2^_p_ > 0.14) [52], and Cohen’s d for the pairwise differences (small effect: 0.2 ≤ d < 0.5; moderate effect: 0.5 ≤ d < 0.8; large effect: d ≥ 0.8) [53]. Additionally, a 90% confidence interval (CI) for η^2^_p_ and 95% CI for each comparison were performed [54]. 

## 4. Results

The three-way ANOVA showed a significant main effect of surface (*p* < 0.001, η^2^_p_ = 0.92), vision (*p* < 0.0001, η^2^_p_ = 0.9), and auditory conditions (*p* < 0.0001, η^2^_p_ = 0.67) factors as well as significant surface × vision (*p* < 0.001, η^2^_p_ = 0.48) and surface × vision × auditory condition (*p* < 0.05, η^2^_p_ = 0.15) interactions on the VmCOP values with a large effect size (Table 1). However, no significant surface × auditory condition or vision × auditory condition interactions were found.

Concerning the auditory condition factor, the post hoc analysis showed that the VmCOP values were significantly (*p* < 0.05) lower when listening to Mozart’s Jupiter compared to the no music condition in two of the postural conditions (firm surface/EO: (*p* < 0.01; 95% CI: 0.27 to 2.22); foam surface/EC: (*p* < 0.001; 95% CI: 0.48 to 2.44) conditions), but not in the other two conditions (firm surface/EC and the foam surface/EO) (Table 2, Figure 5, Figure 6 and Figure 7). 

Regarding the vision factor, the post hoc analysis revealed that in the EC condition, the VmCOP values were significantly (*p* < 0.001) higher compared to the EO condition, irrespective of the auditory or surface condition (Table 2, Figure 5, Figure 6 and Figure 7). 

For the surface factor, the post hoc analysis showed that the VmCOP values were significantly (*p* < 0.001) higher in the foam surface condition compared to the firm surface condition, no matter what the auditory or the vision condition was (Table 2, Figure 5, Figure 6 and Figure 7).

## 5. Discussion

The results of the present study revealed that listening to music (Mozart’s Symphony No. 41 “Jupiter” in C Major, K. 551, starting with the first movement (“I. Allegro vivace”) from the opening of the symphony) has significant positive effects on postural balance under sensory manipulation in middle-aged women. Although beneficial effects of auditory stimulation with music on postural performance have been evidenced in different populations, to the best of our knowledge, there is still a paucity of data on this topic focusing on healthy middle-aged women. Yet, previous investigations, in line with our findings, have been found in other populations (i.e., healthy subjects [31,32], young adults [23], older adults [33], and patients with Parkinson’s disease and visual impairments [34,35,36]). It has been hypothesized that balance improvements following listening to music could be explained by the interaction between the auditory and equilibrium systems in the peripheral receptors of the inner ear, as well as in the CNS [55,56]. Along with vestibular system activation, music has been revealed to stimulate the lateral premotor and supplementary motor areas [30], which may, in turn, enhance the muscular output of the postural balance system. A previous study argued that auditory stimulations such as music stimulation, via loudspeakers, could provide spatial information on the space surrounding subjects through auditory cues [57]. The authors generally argued for their outcomes in terms of an auditory anchorage effect: the sound sources provide a landmark by conveying spatial information, which may enable subjects to decrease their body sway [57]. However, we cannot rely on this explanation, since it has been found that playing audio via headphones affects participants’ awareness of their surroundings [58]. Another factor that may explain the balance improvement is the affective arousal aspect of music, which is likely to influence both motivational and emotional processing. The activation of the emotional neural-based network that involves the dopaminergic mesolimbic projections to the ventral striatum-intraccumbens nuclei is believed to regulate motivational–incentive reinforcements of general behavior [28,59]. Thus, postural balance improvements in response to music listening could be explained by the emotional reactions that instantly activate the cortical–basal ganglia motor loop. Behavioral evidence for a functional interface between the limbic and motor systems [60] and the anatomical–functional sensorimotor integration of basal ganglia and cortical frontal regions [61,62] further support this explanation.

Although there is a growing body of research on the effects of a music stimulus on postural balance, little is available for the specific music type that we used. Indeed, listening to music, like classical music, that is relaxing and highly pleasant was found to reduce pain and increase functional mobility in fibromyalgia patients [63]. The study argued that music reduces pain in fibromyalgia by means of emotional and cognitive mechanisms and that this music-induced analgesic effect is strong enough to increase the subjects’ functional mobility. Others have suggested that the music type might matter by demonstrating that listening to Mozart’s Jupiter reduced body sway compared to other pieces of music [32]. Additionally, in line with previous investigations, we revealed significant differences between the EO and EC conditions for all postural conditions, whatever the auditory condition was [64,65]. Surprisingly, postural balance continued to significantly increase in the EC when feedback was provided through listening to Mozart’s music (Table 1). However, these results indicate that independently of auditory inputs, no interaction between the visual and auditory systems was found. 

Unlike our findings, a previous study showed that different types of music such as classical, pop, and rock had no effects on either static or dynamic balance in healthy adults [23]. The “stochastic resonance” phenomenon may explain these controversial results. Indeed, Ross and Balasubramaniam hypothesized that the reduction in postural sways in response to auditory stimuli could be explained by the stochastic resonance phenomenon [66]. Such a phenomenon occurs when a sensory signal containing information is subthreshold, which means too weak to be detected and integrated by the CNS. 

Importantly, when standing on an unstable surface (foam surface), our results showed that listing to Mozart’s Jupiter enhanced postural balance only in the EC condition (where the visual inputs were removed). In accordance, when auditory input was provided, Dozza et al. (2007) found the greatest improvement in the subjects’ ability to maintain balance in the foam surface condition [67]. These researchers suggested that when standing on a foam surface and receiving limited visual and somatosensory inputs, subjects benefited most from audio-biofeedback. In fact, the CNS integrates and processes information from the vestibular, visual, and somatosensory systems in order to regulate postural stability.

Furthermore, it is well known that standing still requires cognitive processing, and that the more difficult a postural task is, the more cognitive processing is needed [68]. In combination, it has been revealed that music boosts cognitive functions by activating the brain areas that are mainly connected with emotional processing and higher cognitive processes such as the limbic system and frontal lobes, respectively [69,70]. Therefore, the postural balance enhancement found in our results under such challenging conditions (foam surface/EC) may be explained by the benefits of music on cognitive processing. In accordance, it has been revealed that listening to a particular kind of music (such as Mozart’s music) strengthens the connections between particular brain regions, making it easier to choose and “connect” together all the necessary components of sensory stimuli [39].

Overall, our findings suggested that listening to music, particularly Mozart’s Jupiter, could be an alternative method for postural balance improvements in middle-aged women. Postural balance is crucial for daily life activities. Improving it induces significantly better functional mobility, coordination [71,72], and autonomy. Furthermore, a postural balance improvement in response to music listening may reduce the risk and fear of falls as well as the falling incidence. This would, in turn, make a meaningful impact on the quality of life of middle-aged women, since not only falls but also the fear of falling may affect psychological well-being and social behavior [73]. The results of the present study are likely to have important practical implications. Indeed, the promising results of this study, in relation to the effects of music listening on postural balance among these women, should prompt further investigation of its effects on other parameters related to health promotion. In addition, listening to Mozart’s music could be also recommended for these middle-aged women as an effective modality that could reduce the fall incidence and risks related to aging.

This study has some limits that should be addressed by future investigators. First, since we have suggested that a mood and cognitive aspect may explain the postural balance improvements, future investigators may explore the effect of music on mood status and cognitive performance to confirm our hypothesis. In addition, future evaluations of the long-term effects of music listening would be of interest, since some long-term gains may be more likely to be found. Since our study consisted solely of middle-aged and healthy women, the generalization of the findings is difficult for other people such as older adults, men, and people with a specific disease. Future studies conducted on those populations, especially patients with a balance disorder, are highly required. However, it is established that musical genres can impact the neural mechanisms that are associated with postural balance [74]. Future studies are required to address the effects of different musical genres on balance performance. It would also be interesting to explore the effects of other types of music such as jazz, rock, and pop music to support our findings. Furthermore, since using headphones without producing any sound is not the same as not using headphones, it would be of interest to consider including a no headphones condition in future studies in order to obtain more precise results. Finally, as mentioned above, this study was a single-blind trial in which the experimenters were not informed of the sound that the participant was listening to while performing a postural task. It would be very interesting to carry out a double-blind study in which both experimenters and participants are also not informed of the sound being played.

## 6. Conclusions

Overall, this study highlighted the facilitation effect of a particular excerpt from a music piece, Mozart’s Jupiter, on static balance performance in middle-aged women. Indeed, our main findings indicated that listening to Mozart’s Jupiter significantly improves postural balance among middle-aged women during simple and more challenging postural conditions and under different sensory manipulations (EC and foam surface conditions). These improvements could offer greater potential for everyday functioning, help minimize fall risks and related injuries, and improve their quality of life. Therefore, it is recommended for middle-aged women to integrate Mozart’s music into their routines as an effective strategy for enhancing their static balance during daily life activities. 

Clinicians may, therefore, consider music, particularly Mozart’s Jupiter, while designing intervention strategies for middle-aged women to improve their postural performance while performing daily physical activities and, thus, achieve healthy aging. The use of music may prove to be an inexpensive method of treatment that has great societal implications. Additionally, the effect of Mozart’s music should be investigated further to ascertain why it is different in outcomes when compared to other pieces of music. Further applications of this therapy in controlled studies involving subjects with balance disorders are needed. 

## Figures and Tables

**Figure 1 sensors-24-00202-f001:**
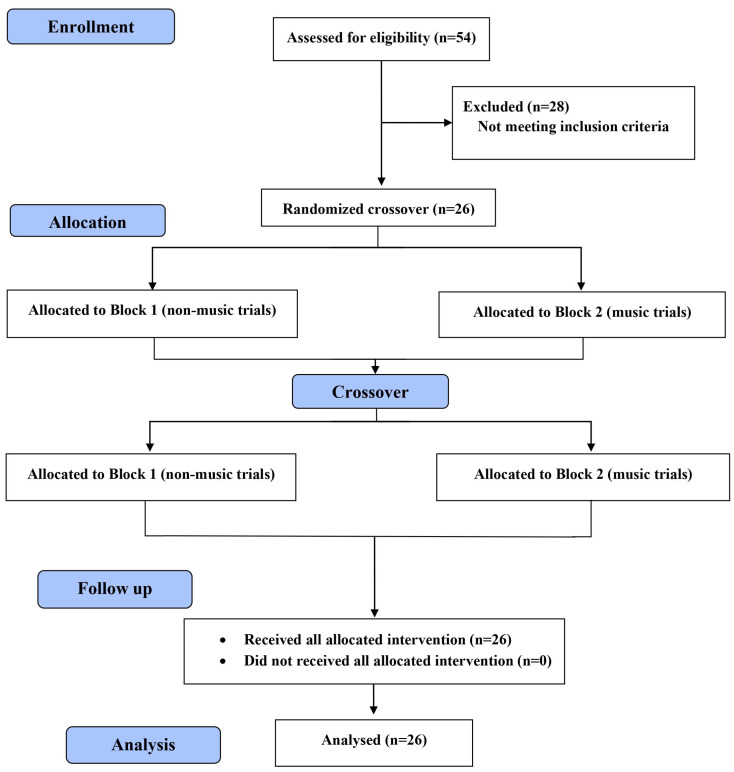
Flow chart of the study.

**Figure 2 sensors-24-00202-f002:**
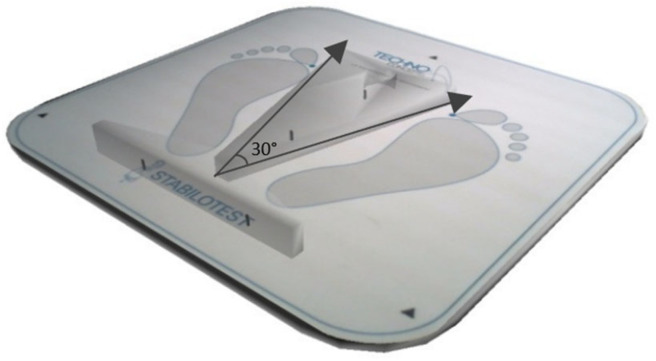
Stabilometric platform (posture Win©).

**Figure 3 sensors-24-00202-f003:**
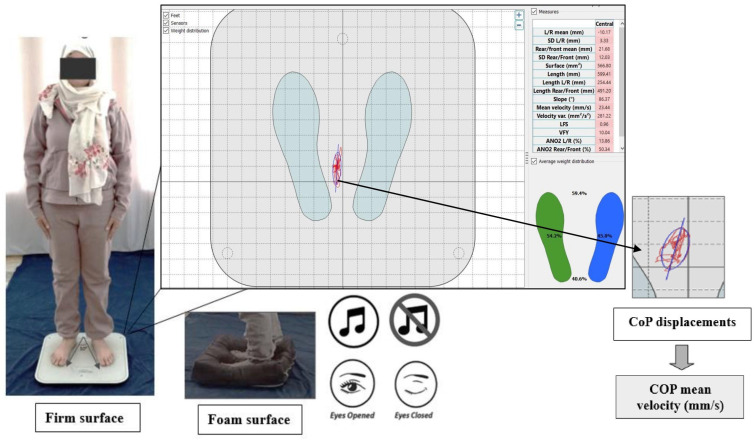
Postural balance assessment.

**Figure 4 sensors-24-00202-f004:**
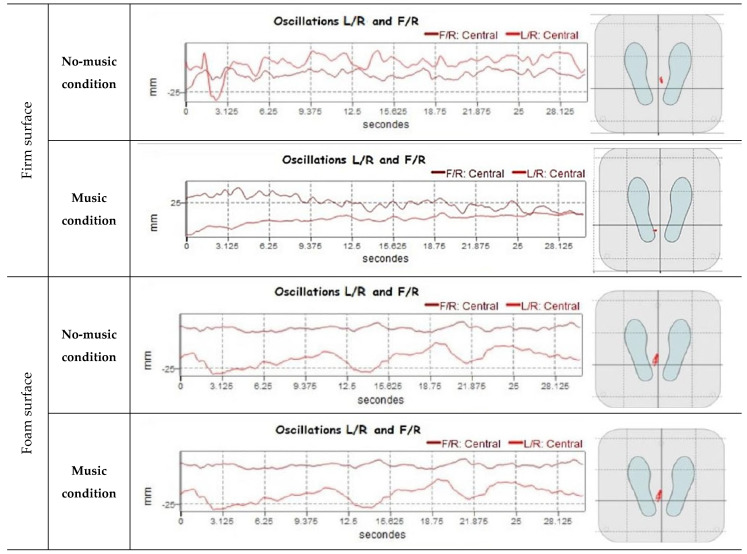
Example of CoP oscillations while listening to music versus no music condition.

**Figure 5 sensors-24-00202-f005:**
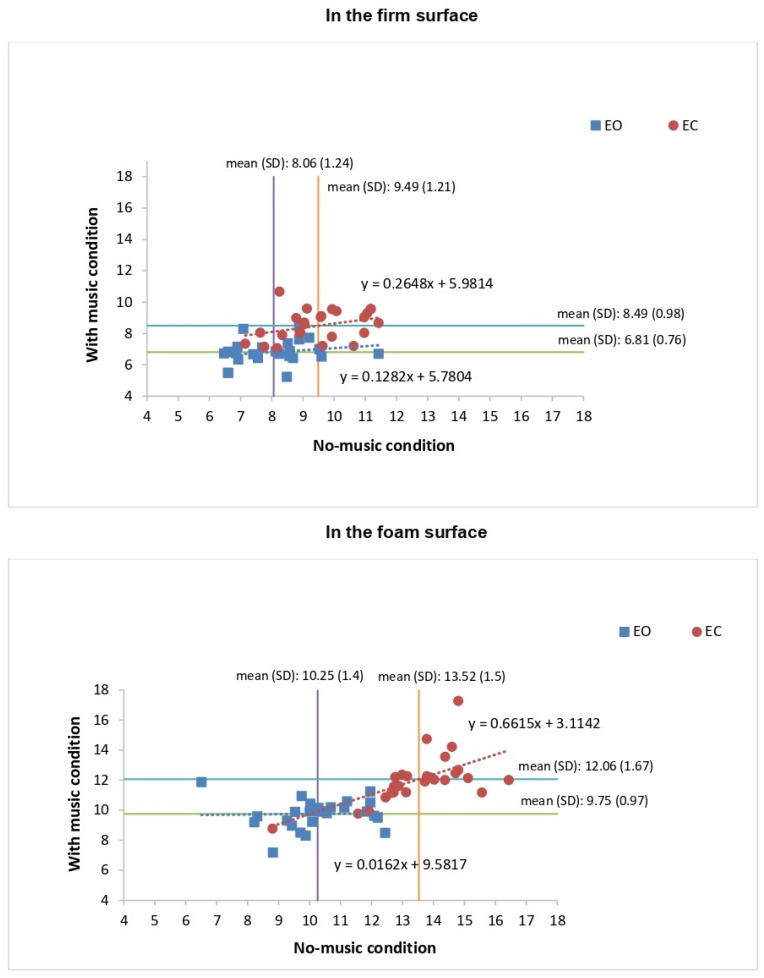
Plots showing the center of pressure mean velocity (VmCOP) values for all participants under the eyes-open (EO) and eyes-closed (EC) conditions on the firm and foam surfaces during two auditory conditions (no music vs. with music condition).

**Figure 6 sensors-24-00202-f006:**
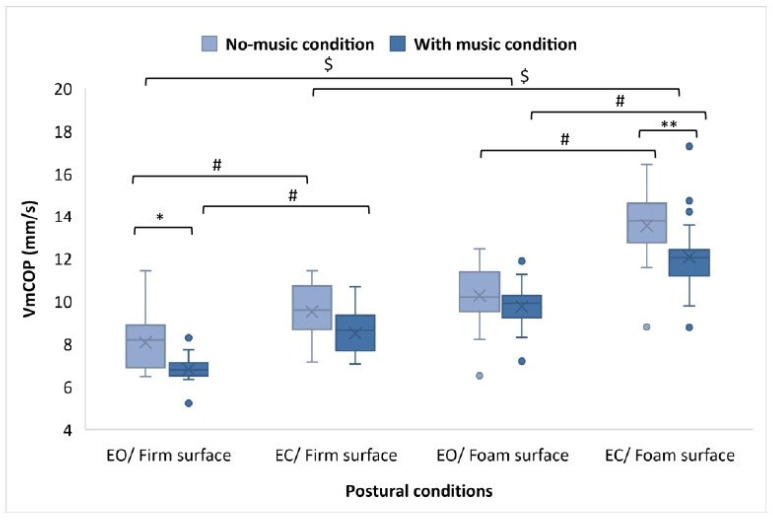
Box-and-whisker plot showing the effect of Mozart’s music on postural balance (the center of pressure mean velocity (VmCOP) values) in both eyes-open (EO) and eyes-closed (EC) conditions on the firm and foam surfaces among middle-aged women. * Significant difference (*p* < 0.01) between no music and Mozart’s Jupiter; ** significant difference at *p* < 0.001 between no music and Mozart’s Jupiter; # significant difference (*p* < 0.001) between EO and EC; $ significant difference (*p* < 0.001) between firm surface and foam surface.

**Figure 7 sensors-24-00202-f007:**
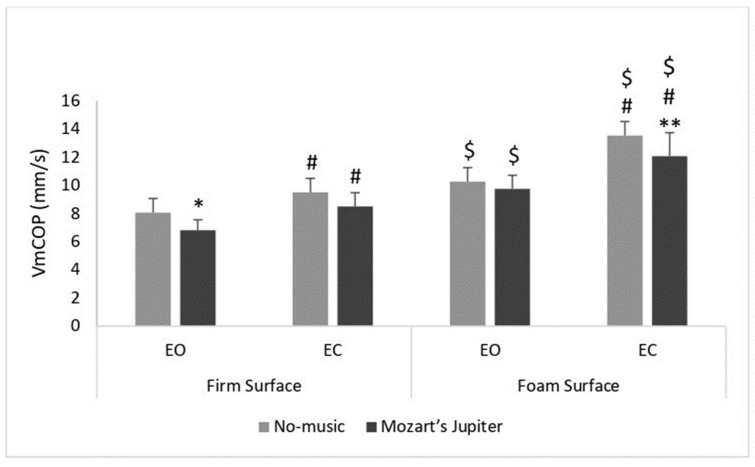
Mean ± SD of the center of pressure mean velocity (VmCOP) parameter during two auditory conditions (no music vs. Mozart’s Jupiter) in both eyes-open (EO) and eyes-closed (EC) conditions on the firm and foam surfaces among middle-aged women. * Significant difference (*p* < 0.01) between no music and Mozart’s Jupiter; ** significant difference at *p* < 0.001; # significant difference (*p* < 0.001) between EO and EC; $ significant difference (*p* < 0.001) between firm surface and foam surface.

**Table 1 sensors-24-00202-t001:** Summary of ANOVA results of the postural balance (center of pressure mean velocity (VmCOP)) parameter showing postural variables’ statistical values (F, *p*, η^2^_p_, 90% confidence interval (CI)) under the eyes-open (EO) and eyes-closed (EC) conditions on the firm and foam surfaces during two auditory conditions (no music vs. Mozart’s Jupiter) in bipedal postures among middle-aged women.

	VmCOP (mm/s)				
	F	*p*-Value	η^2^_p_	90% CI	
				Lower Limit	Upper Limit
Vision Conditions	232.43	<0.001	0.9	0.82	0.93
Surface Conditions	297.51	<0.001	0.92	0.85	0.94
Auditory Conditions	51.19	<0.001	0.67	0.45	0.76
Surface × Vision Conditions	23.96	<0.001	0.48	0.23	0.63
Vision × Auditory Conditions	1.2	0.28	0.04	0	0.21
Surface × Auditory Conditions	0.4	0.52	0.01	0	0.15
Vision × Surface × Auditory Conditions	4.66	<0.05	0.15	0.1	0.35

**Table 2 sensors-24-00202-t002:** Means ± SD of the center of pressure mean velocity (VmCOP) in the eyes-open (EO) and eyes-closed (EC) conditions on the firm and foam surfaces during the two auditory conditions (no music vs. Mozart’s Jupiter) in middle-aged women.

	No Music	Mozart’s Jupiter
Firm Surface		
EO	8.06 ± 1.24	6.81 ± 0.74 *
EC	9.49 ± 1.21 ^#^	8.50 ±0.98 ^#^
Foam Surface		
EO	10.26 ± 1.40 ^$^	9.75 ± 0.98 ^$^
EC	13.53 ± 1.50 ^#$^	12.06 ± 1.65 **^#$^

* Significant difference (*p* < 0.01) between no music and Mozart’s Jupiter; ** significant difference at *p* < 0.001; ^#^ significant difference (*p* < 0.001) between EO and EC; ^$^ significant difference (*p* < 0.001) between firm surface and foam surface.

## Data Availability

The datasets used and/or analyzed during the current study are available from the corresponding author on reasonable request.

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
