# Peer review of "The Effects of Listening to Music on Postural Balance in Middle-Aged Women"

_sensors, 2023, doi:10.3390/s24010202_

Round 1
Reviewer 1 Report
Comments and Suggestions for Authors
The topic is worth of interest, despite the paper shows several shortcomings and it falls below the standards for reporting a RCT.
The authors should carefully take into account the CONSORT statement. Trial design is unclear. The way of participants’ recruitment was not stated, as the type of randomization. Timing and location not declared. Variability in the treatments administration was not controlled. Blinding and harms not reported. Multivariate methods that control better for the biases single factor repeated measure ANOVAs would have been used. A flow-chart which points out the course of the study is absent. Table 1 seems a Figure. Compliance to treatment was not reported. Generalizability and Limitations are poor. Conclusions are unsound and misleading as for the above issues and the absence of a follow-up as well.
I suggest the authors to contact an expert in methodology of research.
So sorry, best regards,
Comments on the Quality of English Language
Extensive editing of English language required
Author Response
Dear reviewer
Thank you for taking the time to review our article. We know how important time is today, for all of us.
Please see the attachment

Reviewer 2 Report
Comments and Suggestions for Authors
This is an interesting and interdisciplinary paper, linking music and audio signals with healthcare and raising the quality of life of individuals. In this case Authors carry out their studies on a group of middle-aged women. As pointed out, ones posture, balance, etc., may be linked with fall rate, and related harmful accidents. However, some modifications could be done in order to further raise its quality.
Suggestions and comments:
1) Provide adequate description of each institution in the Affiliation section.
2) Do reorganize the structure of your manuscript – the Introduction should end with the highlights and novelty of this paper, along with its structure. Later on, prepare a typical [Review of Related Works] section, describing previously published works of other Authors as well as own elaborations. Then, move to the [Materials and Methods].
3) There are several minor editorial and formatting issues, therefore a careful throughout examination would seem necessary.
4) What kind of headphones were utilized in your study? Provide at least principle technical specs.
5) In what file format was the Mozart’s Jupiter presented? Mention about bitrate, number of channels (e.g., mono or stereo), etc. Was it a lossless or lossy audio codec?
6) Moreover, what kind of device was utilized for the playback? Was it a CD player, or maybe MP3/MP4, smartphone, etc.? If it was always one and the same smartphone, provide its brand and motel, etc.
7) In case of fig. 3, what are the units on the X and Y axis?
8) The results describing the difference between with vs without music, are very clear. Yet, Authors could provide additional comments and feedback on their findings.
9) Do extend the Conclusions section, mention about open issues as well as future study directions.
10) Only 12 pages for a research paper is very short, therefore extend this manuscript, in both the theoretical and research parts.
To sum up, this is a good paper, but it deserves to be a very good one. Authors are encouraged to prepare a revised and extended version of their manuscript.
Author Response

(The authors gave the same response as above.)

Reviewer 3 Report
Comments and Suggestions for Authors
I am glad that I got the opportunity to review this work because I believe that such works or research have practical value. It is reflected in the fact that the results of this research can be implemented very easily, bearing in mind the fact that correction or improvement of postural balance in middle-aged women is achieved by listening to Mozart's Jupiter. In this sense, I express my full support for these types of research, which will not remain only in the domain of theory. Generally rather correctly methodologically set.
The abstract is short, clear, and concise.
In the introductory part, the authors clearly and unequivocally clarified the basic terms related to the problem and the goal of the research and supported it with relevant research. It mentions for the first time the music Mozart's Jupiter, which will be used in the research itself. Regarding it, I would recommend the authors explain in a little more detail why and how they chose the music. The paper shows that this music was used in some similar experiments, but I am interested, and I assume future readers will be, whether rock music (for example "Bohemian Rhapsody" by Queen) would have a similar or the same effect. In fact, it is necessary to support the choice of this type of music with arguments or scientific facts.
The sample of respondents is sufficient for drawing certain conclusions. The permission of the ethics committee regarding the research was also clearly presented.
The choice of statistical analysis is in accordance with the research problem. The obtained results are correctly interpreted, and the discussion is supported by corresponding similar research. The obtained conclusion is correctly derived on the basis of well-placed all parts of the work. The literature used is relevant and the presented scientific facts are well-confirmed. Considering that this type of problem with the population of middle-aged women was the first to be investigated by this group of authors, it is clear that it is difficult to find new research on this topic, but it would still be good to strengthen the work with a few more recent similar studies. All in all, I think that the work could be accepted with minimal corrections (a little more detailed description of the choice of music used in the experiment and reinforcement of recent literature). As such, I believe that the work would have a good readership perspective.
Author Response
Dear Reviewer
Thank you for taking the time to review our article. We know how important time is today, for all of us.
Please see the attachment

Round 2
Reviewer 1 Report
Comments and Suggestions for Authors
Dear Authors,
the paper improved satisfactorily. Congratulations.
Best regards,
Comments on the Quality of English Language
Moderate editing of English language required
Author Response
Dear Reviewer 1,
Time is precious for everyone. Thank you for taking the time to review our manuscript and for your comments and advice.
Reviewer 2 Report
Comments and Suggestions for Authors
Authors have prepared a revised version of their initial submission. Currently, the paper is longer, more informative and pleasant to read. Changes made are adequate and properly justified. The number, quality and scope of cited references is sufficient. There are some minor editorial and formatting issues, but they can be easily overcome at a later stage. Therefore, I do recommend this paper to be processed further. The topic is interesting and interdisciplinary, outcomes of this study could help raise the quality of life of numerous people around the world. I also strongly encourage the Authors to continue their studies – do look for further possible improvements in everyday activities, including different age groups, etc.
Author Response
Dear Reviewer 2,
Time is precious for everyone. Thank you for taking the time to review our manuscript and for your comments and advice. We want to touch as many of the aspects you referred to in the future. We also want the ideas from our studies to clarify, help and be applied to increase the quality of life of certain categories of the population.